# Water-immersion finger-wrinkling improves grip efficiency in handling wet objects

Nick J. Davis●*

Department of Psychology, Manchester Metropolitan University, Manchester, United Kingdom

* n.davis@mmu.ac.uk

## Abstract

For most people, immersing their hands in water leads to wrinkling of the skin of the fingertips. This phenomenon is very striking, yet we know little about why it occurs. It has been proposed that the wrinkles act to distribute water away from the contact surfaces of the fingertip, meaning that wet objects can be grasped more readily. This study examined the coordination between the grip force used to hold an object and the load force exerted on it, when participants used dry or wrinkly fingers, or fingers that were wet but not wrinkly. The results showed that wrinkly fingers reduce the grip force needed to grip a wet object, bringing that force in line with what is needed for handling a dry object. The results suggest that enhancing grip force efficiency in watery environments is a possible adaptive reason for the development of wrinkly fingers.

**Data Availability Statement:** All raw and processed data files from this experiment are available at Figshare.com: https://doi.org/10.6084/m9.figshare.14414780.v4. All analyses were conducted in Matlab, and the analysis routine is available in the same repository.

## Introduction

When human hands and feet are immersed in water, over time the skin becomes wrinkled. The wrinkling is mainly confined to the pads of the fingertips and to the toes. Explanations for the wrinkling of the skin include a passive response of the skin to immersion, or an active process that creates the wrinkles for a functional purpose. There is overwhelming evidence that finger-wrinkling is an active process. The small blood vessels of the fingertip constrict, which creates valleys in the skin surface, triggered by water entering sweat pores [1]. Note that a passive explanation would usually assume that water absorbs into the skin, pushing up ridges. This vasoconstriction appears to occur most readily at a temperature of around 40° Celsius, or the temperature of a warm bath [2]. People with autonomic neurological conditions including Parkinson's, cystic fibrosis, congestive heart failure or diabetic neuropathy may show abnormal or asymmetric wrinkling in the affected parts of the body [3–5].

Given that finger-wrinkling is actively maintained, the natural question is why this would happen. It has been suggested that active finger-wrinkling is an adaptation to aid grasping of objects in watery environments. In order to grasp an object, the grip force used to stabilise the object must be enough to balance the load force, which is generated by the mass of the object and is affected by movements of the object, and must take into account the friction of the interface between the fingertips and the object surface [6–8]. Put simply, a wet stone needs to be gripped harder than the same stone when it is dry, as the friction of the contact surface is

**Funding:** The author received no specific funding for this work.

**Competing interests:** The authors have declared that no competing interests exist.

reduced due to the water. Many authors have linked the wrinkling of the fingertips to this grip- and load-force coordination, with the suggestion that the wrinkles act in the same way as the treads on a car tyre, which help to channel water and to provide ridges of drier contact surfaces on the road [9].

If finger wrinkles do indeed aid grasping, we would expect to see this reflected in the grip force used to manipulate an object. Grip and load force are tightly coupled in both static and dynamic grasps. In consciously-initiated movements grip force changes in parallel with the change in load, and slightly precedes it, suggesting a degree of planning of grip force to cope with the changes of load induced by the inertia of the grasped object [6]. Grip force dynamics typically reflect the dynamics of the load, with the rate of change of grip force tracking the developing load force when the load is predictable, and adjust rapidly when the load changes unpredictably [10]. Frictional properties do not appear to be consciously perceivable during passive touch [11], suggesting that effective integration of information into the grasp plan requires either active exploration of the surface, or higher-level information such as recall of previously-sensed information. The influence of prior information in grip force programming leads to perceptual illusions, such as the size-weight illusion [12].

The purpose of the study was to determine if water-induced fingertip wrinkles give an advantage in manipulating a held object with wet hands, compared to when the fingertips are wet but not wrinkled. Specifically, the wrinkles should afford a more efficient grip force, compared to the load induced by manipulating the object. In this study grip and load force was measured in a task where participants gripped an instrument between finger and thumb, and used this to track a load force target as it moved across a screen. It was hypothesised that participants with wrinkly fingers would be more efficient in their grip force than participants with wet but non-wrinkly fingers.

## Materials and methods

### Ethics

All data reported here were collected while the author was in residence at the Science Museum in London, UK. Ethical approval was granted both by the author's then institution, the Department of Psychology, Swansea University, UK, and by the Science Museum. Participants aged 18 or over gave written informed consent to take part in the study, while written parental consent was given in the case of people under 18.

### Participants

After giving informed consent, participants allocated themselves to one of three conditions: these were 'Dry' for people who used dry fingers when taking part, 'Wet' for people who briefly dipped their fingers in water prior to data collection, or 'Wrinkly' for people whose fingers were wrinkled during the experiment. Participants were therefore not blind to their experimental condition, but were naïve to the expected findings. 546 people initially took part in the experiment.

### Procedure

To generate wrinkled fingers, participants immersed their preferred hand in a bath of water kept at 30˚ Celsius, until the fingertips were visibly wrinkled to the satisfaction of the experimenter. To collect grip- and load-force data, two load cells were linked together such that the participant gripped one load cell between the finger and thumb of their preferred hand (Nova-Tech F255, NovaTech Measurements Ltd., UK), and could push or pull the second load cell

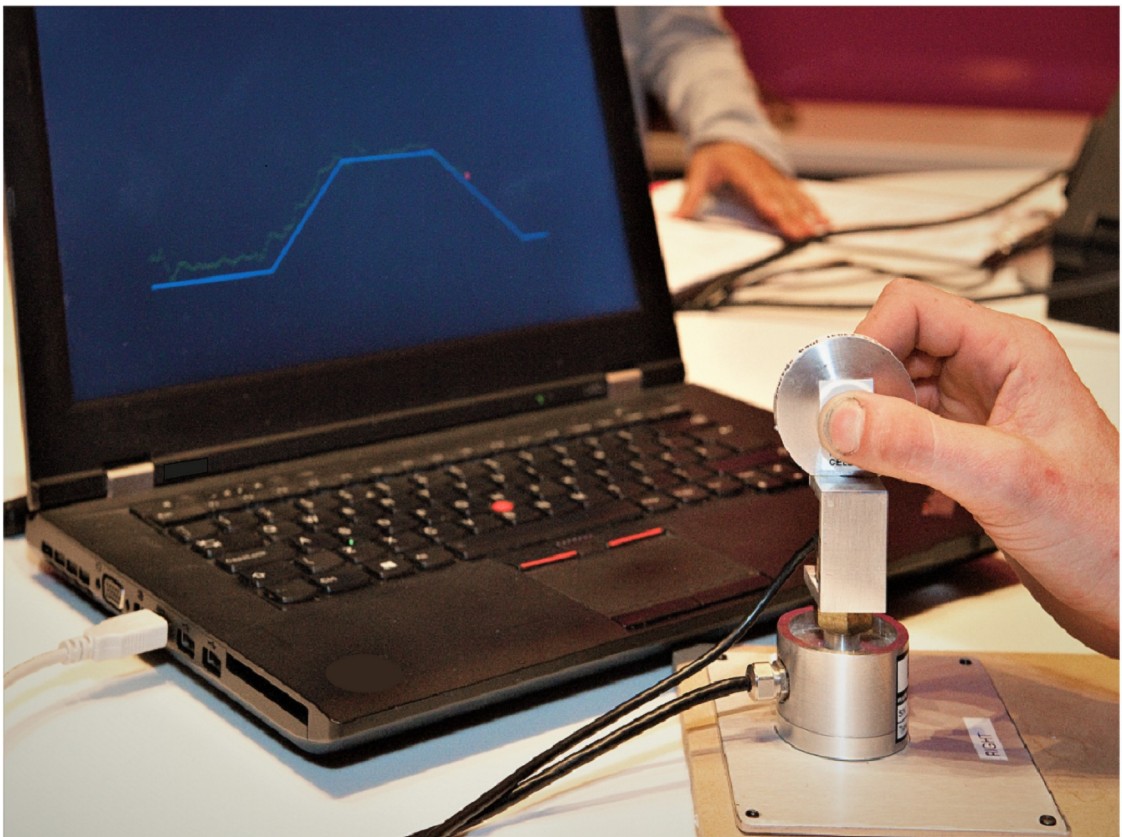

**Fig 1. Picture of the equipment in use.** The participant is gripping a load cell between finger and thumb. The participant's task is to pull up on the second load cell to match a force trace that appears on the laptop monitor. The current load force is shown as a red circle, and the history of the participant's force is shown as a trail of green dots.

vertically (NovaTech F256). The arrangement of the load cells is shown in Fig 1. The load cells were connected to a laptop that displayed the output of the vertical load, using a custom program written in Matlab (The Mathworks, Natick, MA). During a trial, participants were asked to follow a trace that appeared on the screen of the laptop. The target trace appeared as a solid blue line that swept left-to-right across the screen, and the instantaneous output of the vertical load cell was shown as a red circle, with the 'history' of this vertical force shown on the screen as pale dots. Each trial lasted 15 sec. The target trace was static at 0.5 N for 3.5 sec, then rose to 2 N over the course of 3 sec, then was static at 2 N for 4 sec, and dropped to 0.5 N over 3 sec, where it remained for the rest of the trial. Participants each contributed eight trials. Data from the both load cells were digitised at 1000 Hz and stored for later analysis.

## Data analysis

The grip- and load-force data and the target trace were aligned in time, and the load cell data were low-pass filtered with a second-order Butterworth filter set at 20 Hz. Task performance was assessed by determining the correlation between the target force and the load trace, and subjecting these values to a one-way analysis of variance between groups. The primary measure of interest was the ratio of the grip force to the load force. A segment of 3,000 samples (3 sec) was taken from the static phase of the lift. The mean load force and the mean grip force were taken from this time range, and a mean grip:load force ratio was taken for each

participant. These measures were subjected to a one-way analysis of variance, with fingertip condition as a between-subjects factor (Wet, Dry or Wrinkled).

The lag between the change of grip force and the change of load force was also measured, using a cross-correlation between the two traces with a maximum lag of ±150 ms. The rate of change of the grip force trace was assessed by fitting a linear slope to the trace in two windows: between 4 sec and 6 sec into the trial ("attack phase"), and between 11 sec and 13 sec ("decay phase"). These data were also subjected to one-way analysis of variance.

Individual trials were excluded from analysis if the load force trace did not significantly differ from 0 N in the second half of the static hold (suggesting that the participant was not following the target), or if the grip force was more than ten times greater than the load force (suggesting an excessively high grip). A participant was excluded from analysis if more than three trials were excluded based on these criteria.

Statistical analyses were performed within Matlab, with the threshold for statistical significance set at alpha < 0.05. Bartlett's test (Matlab function 'vartestn') was used to test for homogeneity of variance prior to running the analyses of variance. In the one case of violation of this test, the Kruskal-Wallis test was used instead. Post-hoc tests used Matlab's 'multcompare' function, which uses the Tukey-Kramer HSD correction for multiple comparisons after an analysis of variance, or a mean ranks test after the Kruskal-Wallis test had been used.

## Data accessibility

All raw and processed data files from this experiment are available at Figshare.com: https://doi.org/10.6084/m9.figshare.14414780.v4. All analyses were conducted in Matlab, and the analysis routine is available in the same repository.

## Results

After automatic analysis of the force traces, 516 participants' data were analysed. Of these participants, 309 identified as female and 217 as male, and the mean age was 17.7 (SD 13.1). 55 participants chose to use their left hand and 461 their right. 231 participants chose to take part in the Dry condition, 74 in the Wet, and 211 in the Wrinkly condition.

Fig 2 shows the mean traces for the three different conditions. The participants' target force is shown as a black line. The load force traces follow this target line reasonably well, which was expected as the load force was visible to participants as a cursor. The grip force exceeds the load force, as expected. However there is a clear separation between the three traces, with participants with wet fingers using more fingertip force than those who used dry fingers, and with the wrinkly fingers lying between the two.

The correlation between the participants' load force and the target force was good, with no differences between groups [$F_{(2,513)} = 0.953$, $p = 0.386$], and with a mean Pearson's correlation coefficient of 0.628, suggesting that the participants' primary task was executed successfully. A Kruskal-Wallis test was used to test for differences among the conditions in grip:load ratio, as the variances of the groups were not equal according to Bartlett's test ($p = 0.016$). This revealed that the mean of the ratio of grip to load force was different between the conditions [$\chi^2(2) = 24.74$, $p<0.0001$]. Post hoc comparisons found that the ratio was significantly higher for Wet than for Dry ($p<0.0001$) or for Wrinkly ($p = 0.0026$), but that Dry and Wrinkly did not differ from each other ($p = 0.0667$). There was a small but significant correlation of this ratio with the age of the participant [$r(514) = -0.149$, $p<0.001$]. The ratio declined by 0.014 per year of age, although the variance explained by a linear regression was very low ($R^2 = 0.022$). Performing this latter analysis on the conditions separately revealed that a slightly better fit was generated using a logarithmic function, which was significant only for the Dry ($R^2 =$

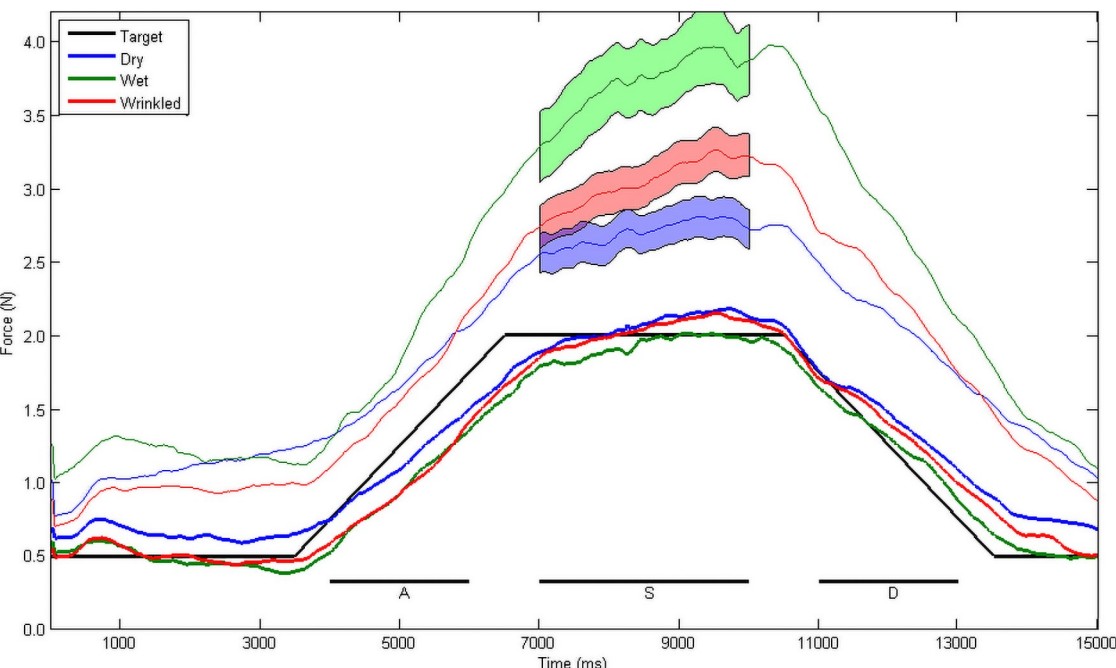

**Fig 2. Comparison of performance across conditions.** Mean grip force (thinner traces) and load force (thicker traces) when participants tracked a load weight target (black line). Participants with wrinkled fingers produced a grip force that did not differ from that used by people with dry fingers in the static hold phase, however people with wet but non-wrinkly fingers used a significantly higher amount of grip. The shaded area indicates the pointwise ±1 standard error of each mean trace. Lines below the trace indicate the attack phase (A) of the trial, the static phase (S) and the decay phase (D).

0.023, p = 0.023) and the Wet ($R^2$ = 0.074, p = 0.021) conditions, but not for Wrinkly. This difference hints at a nonlinear relationship between grip:load ratio and age (I am grateful to an anonymous reviewer for suggesting this analysis).

The lag between the change in grip force and the change in load force was not significantly different between the three groups of participants [$F(2,513) = 0.359$, p = 0.699], but the overall lag was significantly different from simultaneity, with GF leading LF by 22.62 ms. The lag between grip and load force declined significantly with age [$r(514) = -0.338$, p<0.001], with the lead of grip over load change declining by 1.36 ms for each year of age, although the variance explained by a linear regression of these values was low ($R^2$ = 0.114).

The rate of change of grip force was assessed by fitting a simple linear function to the early part of the trial where the target load force was rising (attack phase), and to the later phase where the target was dropping (decay phase). Grip force in the dry condition in the attack phase rose at 0.409 N/sec, which is comparable with the rise of target force of 0.5 N/sec. The slope in the wet condition was significantly greater, at 0.676 N/sec. The slope for the wrinkly condition was intermediate between the two, at 0.501 N/sec, and was not significantly different to either. For the decay phase the grip force trace declined at a rate of -0.345 N/sec, compared to a target of -0.5 N/sec. Participants with wet fingers reduced their force significantly more rapidly, at 0.489 N/sec. People with wrinkly fingers were again intermediate between the two, at 0.690 N/sec, and were significantly different to both wet and dry fingers.

The data for these slope calculations are plotted in Fig 3. It is clear from the boxplots that there is considerable variation in participants' slopes in all conditions. Fig 4 illustrates the relationship between grip and load force by plotting these variables against each other. Fig 4 shows the grand mean trace for all participants within each condition. The target force rises and falls

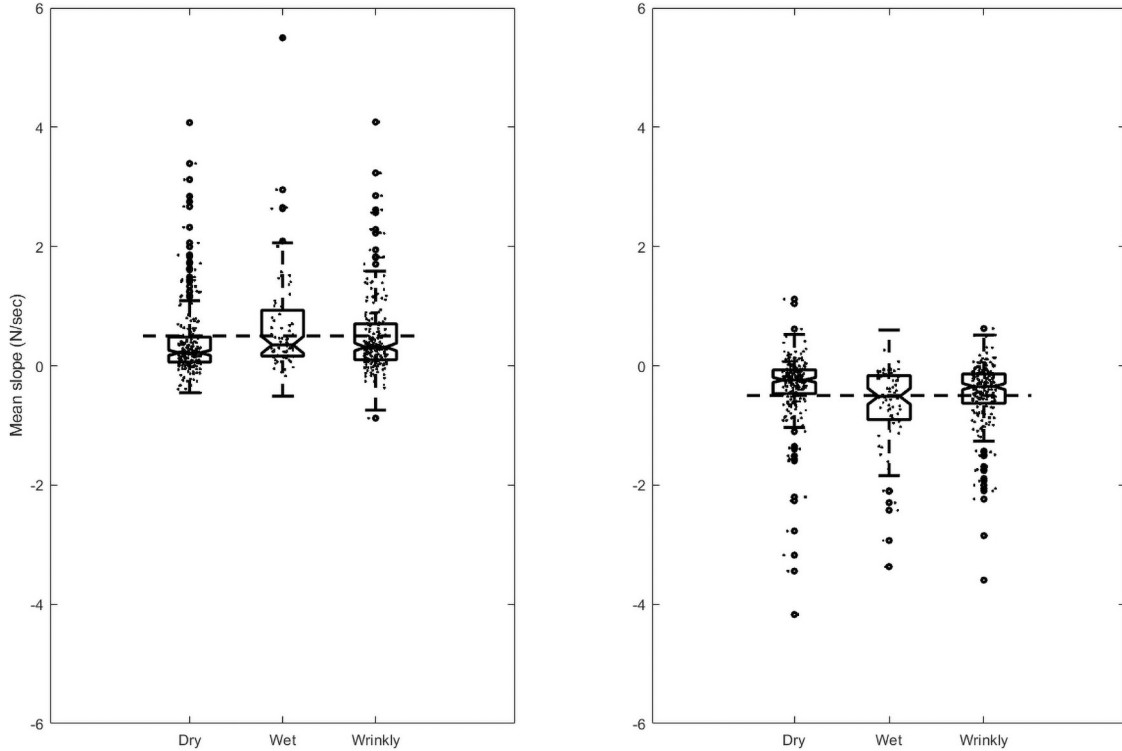

**Fig 3. Rate of change of grip force.** Boxplot of the rate of change of grip force in the 'attack' phase (A) where the target force is rising, and in the 'decay' phase (B) where target force is declining.

linearly between 0.5 and 2.0 N. Participants in the relatively easier Dry condition follow this linear rise and fall reasonably closely, with a slight over-grip (upward shift). The more difficult Wet condition, where the object is more slippery in the grasp, involves a greater safety margin, and a steeper rise in grip force to arrive at a higher margin at the point of highest load force. The Wrinkly condition lies between these two, with a safety margin and rate of change that is intermediate between the Dry and the Wet condition.

## Discussion

There is now converging evidence that finger-wrinkling is an adaptation that aids object manipulation in wet environments [9]. This study has shown that grip efficiency, or the amount by which grip force exceeds the load exerted by the object, is improved when a person has wet and wrinkly fingers, compared to when their fingers are wet but not wrinkly. This ratio of grip force to load force is not significantly different between wrinkly and dry fingers, nor does the relative time difference between the rise of grip force and the rise of load force. Both the grip-to-load ratio and the time difference correlated weakly but significantly with the participants' age. The rate of increase and decrease of grip force was low for dry fingers and high for wet fingers, and for wrinkly fingers was intermediate between the two.

Grip and load force coordination is an important aspect of object handling. The ability to generate the correct amount of grip force for a given load means that the minimum necessary amount of energy is used by the muscles that control the fingers and hands, and means that objects are less likely to be dropped or to be crushed. Efficient grip force coordination is seen in many extant primates, and is likely to have evolved early in the primate lineage [13]. The

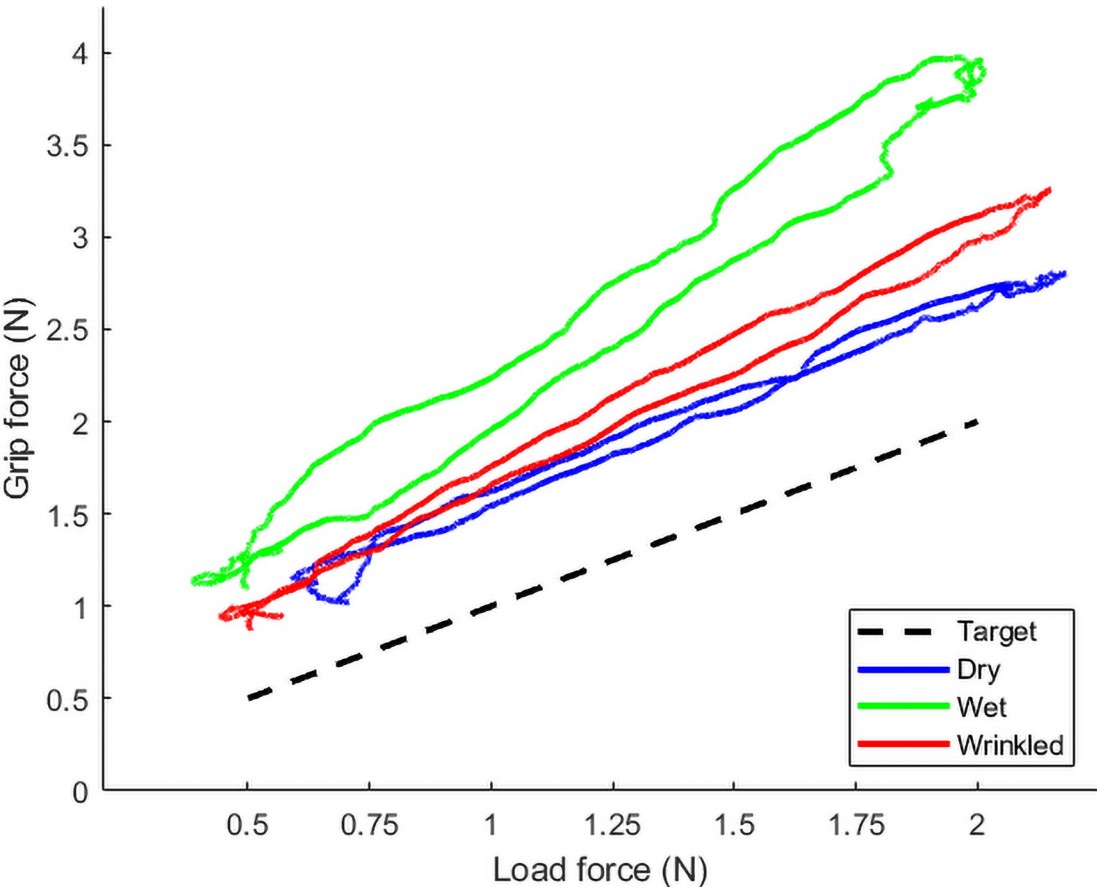

**Fig 4. Relationship between grip and load force in Dry, Wet and Wrinkly conditions.** This illustrates the grand mean of the grip and load forces for the whole duration of the trail, minus the first 1000 ms. The target force is shown as a dashed line. The three grip force traces lie above this line, indicating the safety margin. The 'easiest' condition, Dry (blue trace) follows the target force most closely. The 'hardest', Wet (green trace), shows a higher safety margin, and looser coordination. Participants with Wrinkly fingers (red trace) lie between the two.

grip force required to stabilise a wet object is greater than the force required for a dry object, since the coefficient of friction of the digit-object interface is reduced [8]. It would therefore benefit an animal to gain an advantage in handling wet objects, as this would increase success in hunting and foraging in watery environments. The skin of the fingertip is already adapted for regulation of moisture at the contact surface [14]. Fingertip wrinkles would seem to afford an enhanced advantage in object handling, and may plausibly aid travel and clambering in wet areas, especially if combined with wrinkled toes.

A previous study of object manipulation with wrinkled fingers found that wet objects were moved more quickly when the fingers were wrinkly compared to dry [15]. Importantly, there is no difference in tactile sensitivity in wrinkled fingers compared to dry [16], meaning that people are not trading off acuity for friction at the fingertip. It is therefore reasonable to wonder why healthy people do not have permanently wrinkled fingers. The answer presumably lies in the changes in the mechanical properties of the finger tissues, where there may be lower shear resistance when the finger is wrinkled [17]. Previous studies have also suggested differences in manipulation across the lifespan [18–20]; the present results show age-related effects, although they are rather weak in this sample. Our journey through life leads us to develop strategies for handling familiar and unfamiliar objects, so it seems likely that strategic changes,

along with sensory and motor changes, will affect how children and adults perform tasks with handheld objects [21].

The results presented here should be read in the context of the experiment itself. The age distribution in this sample was rather low, reflecting the public engagement setting of the data collection. Although the effects of age in the data reported here were very small, they were nevertheless statistically significant, so this should be taken into account when comparing these results with others. There may also be effects on performance from inter-individual differences in hand size, in levels of subcutaneous fat, or in lifestyle or genetic factors that were not measured here. Indeed the public environment made it impractical to measure fingertip sensitivity to static sensations or to slips, which may affect individual grip strategies. Finally the experiment only tested one target force pattern and one fingertip contact surface; it is likely that changing the dynamics of the load and the properties of the object would affect grip force coordination [6,22]. Future studies should consider the effect of finger wrinkles on unpredictably changing loads, as the present results do not show a clear separation between the conditions on grip force dynamics. An example of a situation where slippery objects move dynamically would be when a person hunts fish by hand; hand-fishing has been important in low-technology cultures into current times [23], and a fish-rich diet is hypothesised to have promoted the rapid increase in brain size during hominid evolution [24].

In summary, this experiment investigated fine motor coordination when the fingers are affected by water-induced finger-wrinkling. Finger-wrinkling improves grip force coordination when compared to fingers that are wet but not wrinkly, and brings the performance to a level comparable with dry fingers. These results help to explain why humans and their close primate relatives may have developed finger-wrinkling as an adaptation to aid locomotion and foraging in wet environments.

## Acknowledgments

The author is grateful to the staff of the Science Museum, London, for access to the Live Science gallery. Particular thanks are due to Georgie Ariaratnam, and to the many visitors who took part in, or discussed, the experiment.

## Author Contributions

**Conceptualization:** Nick J. Davis.

**Data curation:** Nick J. Davis.

**Formal analysis:** Nick J. Davis.

**Investigation:** Nick J. Davis.

**Methodology:** Nick J. Davis.

**Project administration:** Nick J. Davis.

**Resources:** Nick J. Davis.

**Software:** Nick J. Davis.

**Validation:** Nick J. Davis.

**Visualization:** Nick J. Davis.

**Writing – original draft:** Nick J. Davis.

**Writing – review & editing:** Nick J. Davis.

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
