## [Decision Letter · Decision Letter 0]

19 Mar 2021

PONE-D-21-04649

Water-immersion finger-wrinkling improves grip efficiency in handling wet objects

PLOS ONE

Dear Dr. Davis,

Thank you for submitting your manuscript to PLOS ONE. After careful consideration, we feel that it has merit but does not fully meet PLOS ONE’s publication criteria as it currently stands. Therefore, we invite you to submit a revised version of the manuscript that addresses the points raised during the review process.

Your manuscript has been seen by two expert reviewers. Their comments are attached. Reviewer 2 was entirely positive. Reviewer 1 had some major reservations. I invite you submit a major revision that addresses all comments of Reviewer 1. These include especially the framing of the research question, the description of the methods and results, and the analysis of age groups.

We look forward to receiving your revised manuscript.

Kind regards,

Markus Lappe

Academic Editor

PLOS ONE

Journal Requirements:

Reviewers' comments:

Reviewer's Responses to Questions

**Comments to the Author**

1. Is the manuscript technically sound, and do the data support the conclusions?

Reviewer #1: Partly

Reviewer #2: Yes

2. Has the statistical analysis been performed appropriately and rigorously? 

Reviewer #1: No

Reviewer #2: Yes

3. Have the authors made all data underlying the findings in their manuscript fully available?

Reviewer #1: Yes

Reviewer #2: Yes

4. Is the manuscript presented in an intelligible fashion and written in standard English?

Reviewer #1: Yes

Reviewer #2: Yes

5. Review Comments to the Author

Reviewer #1: The manuscript deals with the relationship between grip force and load force of a simple finger force movement under three conditions: dry fingers (reference condition), wet fingers and wrinkling fingers. The author shows that in the wet condition, a significantly higher grip force is developed in the load task than in the dry condition. Interestingly, the wrinkling fingers condition results in a higher grip force than the reference condition, but lower than with wet fingers.

The work is of interest because of the number of participants analysed (N=516). Nevertheless, I have several concerns about the manuscript that need detailed and thoroughly revision. The most important are the following:

1) The research question(s) is not precisely stated nor justified.

2) The paper looks at a very heterogeneous age group. Considering the data provided by the author, the range is 2 - 71 years. This is neither in anatomical-physiological nor in psychological-cognitive sense a uniform group from whose behaviour one can draw general conclusions. Considering the age range, the assumption of a linear relationship between grip/load ratio and age is worth considering.

3) Why were characteristics of touch sensitivity not measured? There is psycho-physical and anatomical evidence for age-related differences.

4) The description of the methodology is insufficient. Which statistical tests were used, which post-hoc tests, which correction procedures for unequal numbers of participants (e.g. Dry=231, Wet=72, Wrinkly=211)?

The description of the figures is insufficient, the legends are incomplete and contain commented parts that belong in the discussion. What is the meaning of the coloured areas in Fig. 2? How was the average calculated? Is it the mean of all valid trials or is it the mean of the subjects' individual averages? In the latter case, how was intra-individual variability considered when calculating the averages (see age distribution)?

In which interval were the data for Fig. 4 calculated? What does the inter-individual variability look like?

Reviewer #2: This is a simple and clear experiment performed on a high number of individuals. It is aimed at describing the usefulness of finger wrinkles evoked by hand immersion in water.

The study shows that wrinkled fingers exert better grip force coordination than wet fingers without wrinkles. The performance with wrinkles is closest to dry fingers than to wet and wrinkleless fingers. Wrinkly fingers reduce the grip force required to grasp objects, indeed participants with wet fingers used more fingertip force than participants with dry fingers, while participants with wrinkly fingers lied in between. Evolutionary considerations in the discussion open the field to different scientific domains.

Although I see some limitations of the study, most of them acknowledged in the manuscript, I think this work is fresh and interesting. A positive aspect is the number of participants (more than 500). A nice evolution along this line would be to test the same task in the same individuals in the 3 different experimental situations, rather than 3 groups of participants, each one involved in only one task.

6. PLOS authors have the option to publish the peer review history of their article (what does this mean?). If published, this will include your full peer review and any attached files.

Reviewer #1: No

Reviewer #2: No

---

## [Author Response · Author response to Decision Letter 0]

14 Apr 2021

Reviewer #1

The reviewer makes some excellent points, which are both detailed and thoughtful. I am very grateful for these comments, which I think have led to an improved manuscript. Where I have made additions to the manuscript in response to Reviewer #1’s comments, I have highlighted these in yellow in the revised manuscript. Note that I have also updated the analysis script that is filed with the data in the online repository.

1) The research question(s) is not precisely stated nor justified.

I agree that the research question is not completely clear. In the final paragraph of the Introduction I have clarified the purpose of the experiment. I believe that the experiment is well-justified in the preceding paragraphs, where I highlight the state of existing knowledge about wrinkled fingers, and touch on the relevant findings in grip- and load-force coordination. I had also planned for this work to speak to the position of skilled motor action in the evolution of the human lineage (as suggested in the Changizi “rain treads” paper), however I now realise that the supporting evidence on finger-wrinkles in non-human relatives is not available.

2) The paper looks at a very heterogeneous age group. Considering the data provided by the author, the range is 2 - 71 years. This is neither in anatomical-physiological nor in psychological-cognitive sense a uniform group from whose behaviour one can draw general conclusions. Considering the age range, the assumption of a linear relationship between grip/load ratio and age is worth considering.

I agree that the data here covers a wide range of ages, as well as other relevant demographics (as acknowledged in the Discussion). It is certainly true that grip- and load-force coordination changes through the lifespan, and I also touched on this (briefly) in the Discussion. The reviewer makes the very good point that we might not expect a linear relationship to hold between these measures when looking at such a diverse cohort. However I did not have any reason to hypothesise any other form of relationship either, so I planned to use the simplest possible analysis to capture what I suspected would be the linear change in the middle years. It turns out that the age effects were rather small, and not very surprising (young children were worse, but not much worse). However I included the analysis for completeness. This analysis motivates future work to look at developmental differences in wrinkly-finger object manipulation, where the age variable might become categorical.

3) Why were characteristics of touch sensitivity not measured? There is psycho-physical and anatomical evidence for age-related differences.

This is an excellent point, and I agree entirely. The busy environment of the Science Museum meant that a bit of experimental care had to be sacrificed in favour of creating an engaging experience for museum visitors. I plan to conduct follow-up experiments in the lab that would afford this sort of measurement. I hope I have been open in the manuscript about the constraints of the public environment. I have added to the ‘limitations’ section of the Discussion to highlight that static and dynamic touch perception was not measured.

My stint at the Science Museum coincided with a primary and secondary school vacation. I was told that on my busiest day at the museum, 21,000 people passed through the doors. Not all of these people visited my area, but it did make for a very busy few weeks of data collection. If I were lucky enough to do something like this again, I would plan better for an experience that required less hands-on supervision from me, and more self-directed tasks for the visitors.

4) The description of the methodology is insufficient. Which statistical tests were used, which post-hoc tests, which correction procedures for unequal numbers of participants (e.g. Dry=231, Wet=72, Wrinkly=211)?

The primary statistical tests are described in the “Data analysis” section of the Methods, and I believe I have described all primary tests here. I am very grateful that the reviewer asked about the unequal group sizes, as it had not occurred to me to test. I have now used Bartlett’s test to test for homogeneity of variance in the Anovas. Only one test failed this assumption, so a Kruskal-Wallis test was used instead. Post-hoc tests used Matlab’s ‘multcompare’ function, which uses the Tukey-Cramer HSD correction after an Anova, or mean ranks after the Kruskal-Wallis test. I have added this information to the Methods section, under “Data analysis”.

The description of the figures is insufficient, the legends are incomplete and contain commented parts that belong in the discussion. What is the meaning of the coloured areas in Fig. 2? How was the average calculated? Is it the mean of all valid trials or is it the mean of the subjects' individual averages? In the latter case, how was intra-individual variability considered when calculating the averages (see age distribution)?

The description of the shaded area was not clear. It represents the ±1 standard error of the mean grip force at each timepoint in the static hold phase. To calculate this, each participant’s mean grip force was calculated, then the means of these means were calculated to form the traces that are illustrated. The SEM is shown for the static phase only, as this is the part that most directly answers the question I started with. The caption for Figure 2 has been updated to reflect this, and I have added an indication of the three phases of the trial (attack, static, decay).

Intra-individual variance was not analysed in this study, as each person only contributed eight trials. Individual trials were excluded if certain criteria were not met, and individual participants’ data were only included if they made five successful trials. So the analysis only proceeded on ‘valid’ trials. These criteria are stated in the “Data analysis” section of the Methods, and operate automatically within the Matlab analysis script.

In which interval were the data for Fig. 4 calculated? What does the inter-individual variability look like?

This information was missing from the manuscript. It represents the entire trace, minus the first 1000 ms. I have added this information to the caption of Figure 4. This figure is essentially the same as Figure 2, and the variance for the grip trace is shown in Figure 2 for the static phase. I attempted to redraw Figure 4 with the variance illustrated but it turned out to be an unhelpful mess. My intention was for this to look something like a phase plot, where the key information is the principal trajectory through phase space. I don’t think this figure quite achieves that level of helpfulness, however it does show the grip safety margin quite neatly, so I believe it is worth retaining.

Reviewer #2

Reviewer #2 made some generous and positive comments about the manuscript, for which I am very grateful. The reviewer suggests that a within-subjects version of the experiment would be interesting. I entirely agree, especially as grip force strategies tend to be fairly stable within a participant (so revealing differences between conditions). I plan to pursue this as soon as in-person experiments are possible in my region. Again, I thank the reviewer for the heartwarming comments.

---

## [Decision Letter · Decision Letter 1]

18 May 2021

PONE-D-21-04649R1

Water-immersion finger-wrinkling improves grip efficiency in handling wet objects

PLOS ONE

Dear Dr. Davis,

Thank you for submitting your manuscript to PLOS ONE. After careful consideration, we feel that it has merit but does not fully meet PLOS ONE’s publication criteria as it currently stands. Therefore, we invite you to submit a revised version of the manuscript that addresses the points raised during the review process.

Reviewer 2 is happy with your revisions. Reviewer 1 feels that the manuscript does not reflect the full potential of the study with respect to the dependence of the findings on age. Please consider the reviewers comments and decide whether you want to expand the manuscript to include this analysis. As reviewer 1 indicated that s/he is not available for further review on this study that manuscript will not be sent to reviewer 1 again.

We look forward to receiving your revised manuscript.

Kind regards,

Markus Lappe

Academic Editor

PLOS ONE

Journal Requirements:

Reviewers' comments:

Reviewer's Responses to Questions

**Comments to the Author**

1. If the authors have adequately addressed your comments raised in a previous round of review and you feel that this manuscript is now acceptable for publication, you may indicate that here to bypass the “Comments to the Author” section, enter your conflict of interest statement in the “Confidential to Editor” section, and submit your "Accept" recommendation.

Reviewer #1: (No Response)

Reviewer #2: All comments have been addressed

2. Is the manuscript technically sound, and do the data support the conclusions?

Reviewer #1: Partly

Reviewer #2: Yes

3. Has the statistical analysis been performed appropriately and rigorously? 

Reviewer #1: Yes

Reviewer #2: Yes

4. Have the authors made all data underlying the findings in their manuscript fully available?

Reviewer #1: Yes

Reviewer #2: Yes

5. Is the manuscript presented in an intelligible fashion and written in standard English?

Reviewer #1: Yes

Reviewer #2: Yes

6. Review Comments to the Author

Reviewer #1: The author has answered the majority of the comments. Unfortunately, the relevant problem of the age dependence of the grip-force load-force ration has not been answered. The author writes in lines 189-191: " There was a small but significant correlation of this ratio with the age of the participant [r(514)=-0.149, p<0.001]. The ratio declined by 0.014 per year of age, although the variance explained by a linear regression was very low (R2=0.022)." and notes in his reply: "[… ] The reviewer makes the very good point that we might not expect a linear relationship to hold between these measures when looking at such a diverse cohort. However I did not have any reason to hypothesise any other form of relationship either, […]." Here I must clarify that there are clearly other results in the literature. It has been shown that motor skills of different complexities (e.g. Fig. 4.4 in Godde, Voelcker-Rehage, Olk, 2016 or Fig. 2 and 3 in Voelcker-Rehage, 2008) are highly non-linear in their execution as well as their learnability.

I have recalculated the linearity noted by the author using the data provided. The values given by the author (see above) can only be obtained if all data are combined, regardless of the experimental procedure (dry, wet, wrinkled). When the data were considered separately, significant but weak relationships were found for dry and wet and no significant relationship for wrinkled. It can be assumed that the correlation found is due to the high number of participants in the very young to young age range. In order to compensate for the existing numerical imbalance in the number of participants in the age groups, it is necessary to calculate the mean ratios per age group and correlate these values with age. The attached file GripLoadRatio_LogMeanValues.pdf shows my rough calculation of the data. There is a clear non-linear relationship between grip-force load-force ratio and age for all three groups (note the logarithmised y-axis). Only mean values with n>1 were used to calculate the regression curves. These results show that developmental differences can be found even in a simple grasping / loading task. Since this task does not require any complex cognitive-coordinative prerequisites, the observed differences can be assumed to be due to differences in sensory and motor domain. The question therefore arises whether the above-mentioned differences are not only due to cognitive developmental stages but also have a physiological basis. The comparison of the individual regression curves further shows that for the tasks dry and wet a comparable behaviour can be found across the lifespan, whereas under the condition wet higher grip forces are used especially in the younger age groups. This confirms the author's statement that "[…] Fingertip wrinkles would seem to afford an enhanced advantage in object handling, [...] (lines 255-256).

In the present version, the author only confirms the work he himself cites using data collected in a nice "science meets people" setting. The potential of the collected data is not used.

References

Godde, Voelcker-Rehage, Olk. Einführung Gerontopsychologie. UTB Uni-Taschenbücher Bd.4567, Verlag: Ernst Reinhardt / UTB, 2016.

Voelcker-Rehage C. (2008) Motor-skill learning in older adults—a review of studies on age-related differences Eur Rev Aging Phys Act (2008) 5:5–16

Reviewer #2: The Author addressed properly the suggested revisions . I think the results are informative and deserve publication.

7. PLOS authors have the option to publish the peer review history of their article (what does this mean?). If published, this will include your full peer review and any attached files.

Reviewer #1: No

Reviewer #2: No

---

## [Author Response · Author response to Decision Letter 1]

26 May 2021

Reviewer #2 made a suggestion to disaggregate the different conditions (dry, wet, wrinkly) before regressing grip:load ratio against age. Reviewer #2 pointed out a possible nonlinear shape some of these regressions. I ran a similar analysis (using all data, which the reviewer did not do), and found qualitatively similar results. I have added a short amount of text to the Results section and to the Discussion that deals with this nonlinearity, and added one of the works cited in the previous review. As this additional analysis was exploratory I have not dwelt on it, but have acknowledged that there is some interesting work in this area.

---

## [Editor Report · Decision Letter 2]

31 May 2021

Water-immersion finger-wrinkling improves grip efficiency in handling wet objects

PONE-D-21-04649R2

Dear Dr. Davis,

We’re pleased to inform you that your manuscript has been judged scientifically suitable for publication and will be formally accepted for publication once it meets all outstanding technical requirements.

Kind regards,

Markus Lappe

Academic Editor

PLOS ONE
---

## [Editor Report · Acceptance letter]

24 Jun 2021

PONE-D-21-04649R2 

Water-immersion finger-wrinkling improves grip efficiency in handling wet objects 

Dear Dr. Davis:

I'm pleased to inform you that your manuscript has been deemed suitable for publication in PLOS ONE. Congratulations! Your manuscript is now with our production department. 

Kind regards, 

on behalf of

Dr. Markus Lappe 

Academic Editor

PLOS ONE